# SCHREIER-COSET GRAPH REWIRING

## ABSTRACT

Graph Neural Networks (GNNs) provide a principled framework for learning on graph-structured data, yet their expressiveness is fundamentally limited by over-squashing-the exponential compression of information from distant nodes into fixed size vectors. While graph rewiring methods attempt to alleviate this issue by modifying topology, existing approaches can introduce prohibitive computational bottlenecks. We propose Schreier-Coset Graph Rewiring (SCGR), a group-theoretic rewiring method that augments the input graph with a Schreier-coset graph derived from a special linear group $\mathrm{SL}(2, \mathbb{Z}_n)$. Unlike heuristic rewiring, SCGR provides *provable* theoretical guarantees: the auxiliary graph exhibits a spectral gap and a bounded effective resistance, creating low-resistance bypasses for long-range communication. By coupling these two graphs with strength, we ensure that effective resistance between any node pair is bounded, directly mitigating over-squashing. Empirical evaluations demonstrate that SCGR reduces effective resistance by 15-40% across benchmark datasets while maintaining competitive accuracy and lower computational overhead, making it practical for both large-scale and diverse applications.

## 1 INTRODUCTION

Graph Neural Networks (GNNs) are designed to process data exhibiting a graph structure (Hamilton et al., 2017). Their versatility has led to widespread adoption and empirical success across diverse domains (Wu et al., 2020; Abadal et al., 2021). Several GNN variants have emerged. The Graph Convolutional Networks (GCN) (Kipf & Welling, 2016) employs a localized, first-order approximation of spectral graph convolutions. It aggregates normalized features from neighboring nodes to update node embeddings, achieving a computational complexity that scales linearly with the number of edges denoted as $O(E)$. The Graph Isomorphism Network (GIN) (Xu et al., 2018) utilizes sum aggregation neighbor features, followed by a multi-layer perceptron (MLP), to maximize its ability to distinguish between different graph structures. When its MLPs process sufficient capacity, GIN's discriminative power is equivalent to Weisfeiler-Lehman test for graph isomorphism (Huang & Villar, 2021).

Most contemporary GNNs operate under Message Passing Neural Network (MPNN) paradigm (He et al., 2023). In this framework, nodes iteratively exchange information with their neighbors to refine their representations. While more layers are often necessary to capture long-range interactions within the graph, increasing network depth can lead to challenges. Specifically, the receptive field of nodes grows exponentially with depth. This results in large amounts of information from extensive neighborhoods being compressed into fixed-size embeddings (Wilson et al., 2024). This phenomenon, known as over-squashing (Alon & Yahav, 2020), can cause significant information loss (Shi et al., 2023) and thereby substantially limit the expressive capacity of GNNs (Di Giovanni et al., 2023).

Further the performance and behavior of GNNs are intrinsically linked to the underlying graph topology. For instance, the Jacobian of node features is influenced by topological properties such as graph curvature and effective resistance (Di Giovanni et al., 2023; Topping et al., 2021; Black et al., 2023). Various methods are employed to address over-squashing in graph neural networks. *Graph rewiring* techniques by Deac et al. (2022), constructs expander graphs, including Cayley graphs (Wilson et al., 2024) to aid propagation. Wilson et al. (2024) modifies topology using properties such as curvature (Fesser & Weber, 2024), spectral expansion (Karhadkar et al., 2022; Banerjee et al., 2022), and effective resistance (Black et al., 2023) to optimize the flow of information.

*Feature Augmentation* offers an alternative approach. Laplacian Positional Encoding (LapPE) by Dwivedi et al. (2021) injects long-range structural context into node features, reducing the need for deep message-passing layers. However, its $O(n^3)$ eigenvector computation limits scalability and makes it sensitive to topological perturbations. Another method, shortest-path distance encoding directly inputs hop counts (all-pair computation is $O(n(E))$, bypassing intermediate message propagation. This method typically encodes only scalar distances, thus neglecting information about other possible paths and connectivity issues.

In this work, we introduce **Schreier-Coset Graph Rewiring (SCGR)**, a novel framework that augments input graphs with Schrier-coset graphs derived from $SL(2, \mathbb{Z}_n)$. Unlike prior approaches, that rely solely on Cayley expanders or heavy rewiring, *SCGR* provides a principled alternative. Our main contribution are:

- **Formalization of Schreier–coset rewiring.** We define the construction of Schreier–coset graphs and their integration into GNNs as rewiring augmentations. Vertices correspond to cosets of $SL(2, \mathbb{Z}_n)$ modulo an upper-triangular subgroup, with constant-side generators yielding $d$-regular graphs.

- **Theoretical analysis.** We provide rigorous theoretical analysis of *SCGR*, through the analysis of spectral properties, and bounds on effective resistance of the Schreier graph, and the resulting performance guarantees of *SCGR* including over-squashing mitigation in GNNs.

- **Empirical Validation.** We evaluate *SCGR* on benchmark datasets for node and graph classification, as well as synthetic stochastic block models with varying modularity. Results demonstrate that *SCGR* consistently matches, attains higher scores against rewiring baselines, and performs well with varying graph modularity.

## 2 Related Works and Existing Approaches

A common strategy to mitigate structural bottlenecks is to decouple the input graph from the computational graph. Alon & Yahav (2020) proposed rewiring by making the final GNN layer fully adjacent, enabling all nodes to interact directly without full-graph pre-analysis. Graph Transformers (Ying et al., 2021; Kreuzer et al., 2021) follow a similar principle with full connectivity in each layer, but their $O(|V|^2)$ edge complexity restricts scalability. Alternatively, Gilmer et al. (2017) introduced a controller node connected to all others, reducing diameter to 2 with only $O(|V|)$ edges, but risking a new bottleneck by over-centralizing flow.

In *Feature Augmentation*, node/edge attributes are enriched with global signals. Eliasof et al. (2023) concatenate top-$k$ Laplacian eigenvectors to each node, so long-range information need not propagate hop by hop. However, eigen-decomposition costs $O(n^3)$ and $O(nk)$ memory, and suffers from sign ambiguity and batch inefficiency. In *Graph Rewiring*, the input graph is modified to improve spectral properties. Karhadkar et al. (2022) and others (Banerjee et al., 2022; Black et al., 2023; Arnaiz-Rodríguez et al., 2022) use spectral metrics or effective resistance to reduce oversquashing. While effective, these approaches demand costly full-graph analysis. In *Expander Graphs*-Expanders provide favorable spectral gap and resistance. Banerjee et al. (2022) proposed random local rewiring inspired by expanders. Shirzad et al. (2023) combined expanders with virtual nodes in a graph transformer. Wilson et al. (2024) extended this to Cayley propagation, inflating node count to $O(n^3)$ and requiring heavy padding/truncation.

## 3 Schreier-Coset Graphs Rewiring (SCGR) for GNNs

### 3.1 Preliminaries

**Graphs.** Let $G = (V, E)$ denote an undirected, connected, and non-bipartite graph with node set $V$ and edge set $E$. Its adjacency matrix is $A \in \mathbb{R}^{n \times n}$ with entries $A_{ij} = 1$ if $(i, j) \in E$ and 0 otherwise, where $|V| = n$. Define the diagonal degree matrix $D = \text{diag}(d_1, \ldots, d_n)$ with $D_{vv} = d_v$. The normalized Laplacian is

$$L = D^{-1/2}(D - A)D^{-1/2}.$$

The eigenvalues of $L$ satisfy $0 = \lambda_0 \leq \lambda_1 \leq \cdots \leq \lambda_{n-1}$. The eigenvector associated with $\lambda_1$ (the algebraic connectivity) is known as the *Fiedler vector*. It provides a canonical one-dimensional embedding of the nodes that reflects graph connectivity.

**Special Linear Group** $SL(2, \mathbb{Z}_n)$. Let $\mathbb{Z}_n = \mathbb{Z}/n\mathbb{Z}$ denote the ring of integers modulo $n$. The group $\mathcal{G} = SL(2, \mathbb{Z}_n)$ is defined as:

$$\mathcal{G} = SL(2, \mathbb{Z}_n) = \left\{ M \in \mathbb{Z}_n^{2\times 2} \mid \det(M) \equiv 1 \pmod{n} \right\}.$$

Here, $n$ depends on the input graph size.

**Subgroup.** Let $H$ be a subgroup $H \subset SL(2, \mathbb{Z}_n)$ which consists of diagonal matrices with unit determinant within $\mathcal{G}$:

$$H = \left\{ \begin{pmatrix} a & 0 \\ 0 & d \end{pmatrix} \in \mathcal{G} \mid ad \equiv 1 \pmod{n} \right\}.$$

**Generator.** Let $\mathbb{S}$ be the generator set:

$$\mathbb{S} = \left\{ \begin{pmatrix} 1 & \pm 1 \\ 0 & 1 \end{pmatrix}, \begin{pmatrix} 1 & 0 \\ \pm 1 & 1 \end{pmatrix} \right\} \bmod n.$$

**Expander graphs.** An expander graph is sparse yet highly connected, with edges scaling linearly with nodes. We use pre-computed expander graphs based on Cayley graphs $\mathrm{Cay}(\mathcal{G}; \mathbb{S})$ derived from the special linear group $\mathcal{G} = SL(2, \mathbb{Z}_n$ with generating set $\mathbb{S}$. While these graphs have good expansion properties, achieving large node counts is often impractical due to the node count formula:

$$|V(\mathrm{Cay}(\mathcal{G}; \mathbb{S}))| = n^3 \prod_{\text{prime } p|n} \left( 1 - \frac{1}{p^2} \right)$$

which creates excessive memory requirements for large $n$ (where $n$ is the smallest value satisfying $|V(\mathrm{Cay}(SL(2, \mathbb{Z}_n); \mathbb{S}))| \geq |V|$).

### 3.2 SCHREIER-COSET GRAPH $\Gamma$.

Following Schreier (1927), Schreier–coset graphs provide a permutation representation of finitely generated groups on the cosets of a subgroup of $\mathrm{SL}(2, \mathbb{Z}_n)$. The Schreier-coset graph plays a central role in our rewiring scheme, serving as an auxiliary structure that encodes robust expansion and mixing behavior through group-theoretic symmetries.

Formally, for a group $\mathcal{G}$, a subgroup $H \subseteq \mathcal{G}$, a generating set $\mathbb{S} \subseteq \mathcal{G}$, the Schreier-coset graph $\Gamma = (V_\Gamma, E_\Gamma)$ is defined as:

- **Vertex Set :** $V_\Gamma = \{gH : g \in \mathcal{G}\}$ (collection of right cosets).
- **Edge Set :** For each $gH \in V_\Gamma$ and each $s \in \mathbb{S}$, include an undirected edge $\{gH, (gs)H\} \in E_\Gamma$.

This yields a $d$-regular graph with $d = |\mathbb{S}|$, since each coset has one neighbor for every generator.

In constructing the the Schreier graph, we employ a *canonical construction*. That is, $\Gamma$ is constructed over the group $\mathcal{G} = SL(2, \mathbb{Z}_n)$ with subgroup $H$ consisting of diagonal matrices, and use elementary row operations as generators:

$$\mathbb{S} \approx \left\{ \begin{pmatrix} 1 & \pm 1 \\ 0 & 1 \end{pmatrix}, \begin{pmatrix} 1 & 0 \\ \pm 1 & 1 \end{pmatrix} \right\} \bmod n.$$

The resulting Schreier-coset graph has $|V_\Gamma| = \frac{SL(2, \mathbb{Z}_n)}{|H|} = \frac{n(n^2-1)}{\phi(n)}$ vertices, where $\phi(n)$ is Euler's totient function.

### 3.3 SCHREIER-GUIDED GRAPH REWIRING

We augment the input graph $G_{in} = (V_{in}, E_{in})$ using structure-preserving rewiring guided by the Schreier–coset graph $\Gamma$. Central to this procedure is a locality-preserving mapping $\phi : V_{in} \to V_\Gamma$.

**Spectral Mapping Construction .**  Let $\Phi_{in} : V_{in} \rightarrow \mathbb{R}^r$ be spectral embeddings using the $r$ leading eigenvectors of their respective graph Laplacians. We defined the locality-preserving mapping $\phi : V_{in} \rightarrow V_\Gamma$ by solving:

***Case (i)*** $|V_{\text{in}}| \leq |V_\Gamma|$:

$$\min_{\phi : V_{\text{in}} \hookrightarrow V_\Gamma} \sum_{(u,v) \in E_{\text{in}}} \text{dist}_\Gamma(\phi(u), \phi(v))$$

subject to $\|\Phi_\Gamma(\phi(v)) - \Phi_{\text{in}}(v)\|_2$ being small.

***Case (ii)*** $|V_{\text{in}}| > |V_\Gamma|$: Use disjoint copies $\Gamma^{(1)}, \ldots, \Gamma^{(q)}$ or a product $\Gamma \times K_q$ and apply case (i) per block.

The optimization ensures that neighborhoods in $G_{in}$ are mapped to neighborhoods in $\Gamma$, preserving local structure for effective rewiring.

**Rewiring Strategy.**  Edges are added to $G_{in}$ between nodes $u, v \in V_{in}$ if their Schreier images $\phi(u), \phi(v)$ are connected by short paths in $\Gamma$ but far apart in $G_{in}$. This creates an alternative that leverages the expander properties of $\Gamma$. Specifically:

- *Distance Threshold Selection:* Fix a maximum distance $\ell \geq 1$ in $\Gamma$.
- *Schreier Proximity Detection:* For each pair $(u, v)$, compute $\text{dist}_\Gamma(\phi(u), \phi(v))$.
- *Conditional edge addition:* Add $(u, v)$ if:
    1. $(u, v) \notin E_{in}$ (not already connected),
    2. $\text{dist}_\Gamma(\phi(u), \phi(v)) \leq \ell$ (close in Schreier Graph),
    3. $\text{dist}_{G_{in}}(u, v) > \ell$ (distant in original graph).

The rewired graph is

$$G^{rwd} = (V_{in}, E^{rwd}), \quad E^{rwd} = E_{in} \cup \{(u, v) : \text{dist}_\Gamma(\phi(u), \phi(v)) \leq \ell < \text{dist}_{G_{in}}(u, v)\}.$$

Added edges are weighted by

$$w_{uv} = \epsilon \cdot f(\text{dist}_\Gamma(\phi(u), \phi(v))),$$

where $\epsilon > 0$ is a global strength parameter and $f(\cdot)$ is a monotone decreasing function.

SCGR rests on three key pillars: (i) locality preservation through spectral embeddings, (ii) effective resistance reduction via alternative low-resistance pathways, and (iii) quantifiable over-squashing mitigation while maintaining near-linear complexity. We establish formal theoretical guarantees for each component in the next Section. Here we first analyze the computational complexity of the approach.

The practical implementation of SCGR involves several computational components, each with well-defined complexity bounds.

**Graph Construction:** The Schreier-coset graph $\Gamma$ has $V_\Gamma = O(n)$ vertices for $\pmod{n}$ (and $O(n \cdot \text{polylog}(n))$ in general case), with constant degree $d = |\mathbb{S}| = 4$. Its edge set therefore satisfies $|E_\Gamma| = O(|V_\Gamma|)$. Constructing $\Gamma$ via coset representatives and generator multiplications requires $O(|V_\Gamma|)$ group operations, which can be cached once and reused across multiple input graphs.

**Mapping and Rewiring:** The locality-preserving mapping $\phi : V_{in} \rightarrow V_\Gamma$ can be computed using spectral embeddings of dimension $r \ll |V|$. This requires $O(r \cdot |E_{in}|)$ operations via power iteration on the Laplacian. For rewiring, distance queries $dist_\Gamma(\phi(u), \phi(v))$ can be approximated using truncated BFS or landmark-based embeddings, avoiding a quadratic scan over all pairs. Thus the edge addition process runs in $\tilde{O}(|E_{in}| + |V_{in}|) \cdot \deg_{\Gamma(\ell)}$ where $\deg_{\Gamma(\ell)}$ is the number of Schreier neighbors within distance $\ell$.

**Message Passing:** Each GNN layer on the rewired graph requires $O(|E^{rwd}|)$ operations. Since $E^{rwd} = O(E_{in} + V_{in}) \cdot \deg_{\Gamma(\ell)}$ and $deg_{\Gamma(\ell)}$ grows moderately (expander property), the per-layer complexity remains near-linear in the input size.

**Space Complexity:** The node set is unchanged ($|V^{rwd}| = |V_{in}|$). The added edges are at most $O(|V_{in}| \cdot \deg_{\Gamma(\ell)})$, and $\deg_{\Gamma(\ell)} = O(d^\ell)$ with $d = 4$. Thus the memory overhead is tunable via $\ell$ and typically sub-quadratic.

## 4 THEORETICAL PROPERTIES OF SCGR

We first show that the Schreier-coset graph $\Gamma$ is an expander with strong spectral and mixing properties that enable efficient information propagation through low effective resistance paths.

**Lemma 4.1** (Spectral Gap). *The Schreier-coset graph $\Gamma$ has a spectral gap*

$$\gamma = 1 - \lambda_2(P) > 0,$$

*where $P$ is the transition matrix of the random walk on $\Gamma$.*

**Lemma 4.2** (Expander Mixing). *For the random walk matrix $P$ on $\Gamma$ and all $t \geq 0$,*

$$\left| (P^t)_{iv} - \frac{1}{|V_\Gamma|} \right| \leq (1 - \gamma)^t.$$

*If $t \geq \frac{\log(2|V_\Gamma|)}{\gamma}$, then $(P^t)_{iv} \geq \frac{1}{2|V_\Gamma|}$.*

**Lemma 4.3** (Effective Resistance Bound). *For any vertices $u, v \in V_\Gamma$,*

$$R_{\text{eff}}(u, v) \leq \frac{2}{d\gamma},$$

*where $d = |\mathbb{S}|$ is the degree and $\gamma$ is the spectral gap.*

The bounded effective resistance guarantees that any two nodes in the Schreier graph are well-connected, with resistance inversely proportional to the spectral gap $\gamma$. This property is crucial for creating efficient rewiring patterns.

**Bi-Lipschitz Control via Spectral Embeddings.** The spectral alignment between $G_{in}$ and $\Gamma$ preserves distance relationships up to a controlled factor, ensuring that the rewiring preserves meaningful structural relationships.

**Theorem 4.1** (Lipschitz Locality). *If $\Phi_{in}$ and $\Phi_\Gamma$ are bi-Lipschitz on relevant scales, then there exists $c \geq 1$ such that*

$$\text{dist}_\Gamma(\phi(u), \phi(v)) \leq c \cdot \text{dist}_{in}(u, v)$$

*for all $u, v \in V_{in}$.*

**Effective Resistance Analysis of the Rewired Graph.** Theorem 4.2 show that the rewiring process creates alternative pathways between distant nodes, significantly reducing effective resistance and enabling better information flow.

**Theorem 4.2** (Effective Resistance in Rewired Graph). *In the rewired graph $G^{rwd}$, the effective resistance between nodes $u, v \in V_{in}$ satisfies*

$$R_{\text{eff}}^{rwd}(u, v) \leq \min \left\{ R_{\text{eff}}^{in}(u, v), \frac{1}{\epsilon} R_{\text{eff}}^\Gamma(\phi(u), \phi(v)) + \frac{2}{\epsilon} \right\}.$$

*Sketch.* The rewired graph can be viewed as an electric network where current can (i) route entirely in the original $G_{\text{in}}$, which gives $R_{\text{eff}}^{\text{in}}$; (ii) route via the $\Gamma$-layer uses two connectors (each $1/\varepsilon$) in series with a $\Gamma$ path whose energy scales as $R_{\text{eff}}^\Gamma/\varepsilon$; Thomson's principle gives the second term. Rayleigh monotonicity/Kron reduction also imply $R_{\text{eff}}^{\text{rwd}} \leq R_{\text{eff}}^{\text{in}}$.

**Information Flow and Over-Squashing Mitigation.** The connection between effective resistance and information propagation in neural networks is well established. In message-passing networks, the gradient flow between distant nodes is inversely proportional to their effective resistance.

**Theorem 4.3** (Over-Squashing Mitigation). *For nodes $u, v \in V_{in}$ with large distance,*

$$\rho(u, v) = \frac{R_{\text{eff}}^{in}(u, v)}{R_{\text{eff}}^{rwd}(u, v)} \geq \max \left\{ 1, \frac{R_{\text{eff}}^{in}(u, v) \cdot \epsilon}{R_{\text{eff}}^\Gamma(\phi(u), \phi(v)) + 2} \right\}.$$

When $R_{eff}^{in}(u, v)$ grows exponentially with distance, but $R_{eff}^\Gamma(\phi(u), \phi(v)) \leq \frac{2}{d\gamma}$ remains bounded, the improvement factor $\rho(u, v)$ can be exponentially large.

Table 1: Performance comparison of SCGR against baseline models across six standard benchmark datasets.

| Model | Am. Comp. | Am. Photo | CiteS. | Co. CS | Cora | PubMed |
|---|---|---|---|---|---|---|
| LogReg | $0.6410 \pm 0.0570$ | $0.7300 \pm 0.0650$ | - | $0.8640 \pm 0.0900$ | - | - |
| MLP | $0.4490 \pm 0.0580$ | $0.6960 \pm 0.0380$ | $0.5880 \pm 0.0220$ | $0.8830 \pm 0.0070$ | $0.5980 \pm 0.0240$ | $0.7010 \pm 0.0070$ |
| GAT | $0.7800 \pm 0.1900$ | $0.8570 \pm 0.2030$ | $0.6890 \pm 0.0170$ | $0.9050 \pm 0.0060$ | $\mathbf{0.8080 \pm 0.0160}$ | $0.7780 \pm 0.0210$ |
| GCN | $0.8260 \pm 0.0240$ | $0.9120 \pm 0.0120$ | $0.6820 \pm 0.0160$ | $0.9111 \pm 0.0050$ | $0.7910 \pm 0.0180$ | $0.7880 \pm 0.0060$ |
| MoNET | $0.8350 \pm 0.0220$ | $0.9120 \pm 0.0130$ | $\mathbf{0.7120 \pm 0.0020}$ | $0.9080 \pm 0.0600$ | $0.5980 \pm 0.0080$ | $0.7860 \pm 0.0230$ |
| LabelProp | $0.7080 \pm 0.0810$ | $0.7260 \pm 0.0111$ | $0.6780 \pm 0.0210$ | $0.7360 \pm 0.0390$ | $0.5050 \pm 0.0150$ | $0.7050 \pm 0.0530$ |
| LabelProp NL | $0.7500 \pm 0.0390$ | $0.8390 \pm 0.0270$ | $0.6670 \pm 0.0220$ | $0.7600 \pm 0.0140$ | $0.5100 \pm 0.0100$ | $0.7230 \pm 0.0290$ |
| GS-mean | $0.8240 \pm 0.0180$ | $0.9140 \pm 0.0130$ | $0.7160 \pm 0.0190$ | $0.9130 \pm 0.0280$ | $0.5860 \pm 0.0160$ | $0.7740 \pm 0.0220$ |
| GS-maxpool | − | $0.9040 \pm 0.0130$ | $0.6750 \pm 0.0230$ | $0.8500 \pm 0.0110$ | $0.4700 \pm 0.0150$ | $0.7610 \pm 0.0230$ |
| GS-meanpool | $0.8960 \pm 0.0090$ | $0.9070 \pm 0.0160$ | $0.6860 \pm 0.0240$ | $0.8960 \pm 0.00090$ | $0.4050 \pm 0.0150$ | $0.7650 \pm 0.0240$ |
| + SCGR | $\mathbf{0.9031 \pm 0.0062}$ | $\mathbf{0.9400 \pm 0.0026}$ | $0.6180 \pm 0.0215$ | $\mathbf{0.9211 \pm 0.0022}$ | $0.7957 \pm 0.0058$ | $\mathbf{0.7894 \pm 0.0097}$ |

**Performance Guarantees.** Combining the above results, we can establish comprehensive performance bounds for the SCGR approach.

**Theorem 4.4** (Performance Guarantees). *Let $G_{in}$ be an input graph with diameter $D$ and maximum effective resistance $R_{\max}^{in}$. Using Schreier-guided rewiring with a Schreier graph $\Gamma$ of spectral gap $\gamma > 0$, the following holds:*

- *For all $u, v \in V_{in}$: $R_{\mathrm{eff}}^{rwd}(u, v) \leq \min\left\{ R_{\max}^{in}, \frac{1}{\epsilon}\left(\frac{2}{d\gamma} + 2\right) \right\}$.*

- *Information can propagate between any nodes with resistance bounded by $\frac{1}{\epsilon}\left(\frac{2}{d\gamma} + 2\right)$.*

- *The over-squashing factor is reduced by at least $\frac{R_{\max}^{in} \cdot \epsilon d\gamma}{2(d\gamma + 2)}$.*

These results make the impact of SCGR *dimension-free and explicit*: (i) the post-rewiring resistance between *all* node pairs is uniformly bounded by a term depending only on the expander parameters of $\Gamma$ and the coupling $\varepsilon$ (not on the input graph geometry), and (ii) the improvement factor scales as $O\left(\varepsilon\, d\, \gamma \cdot R_{\mathrm{eff}}^{in}\right)$, quantifying how SCGR collapses long-range bottlenecks that cause over-squashing. Crucially, these gains come with *near-linear* overhead: the $\Gamma$-overlay is constant-degree and tunable via $\varepsilon$, delivering provable long-range communication.

*All proofs are provided in the Appendix.*

## 5 EXPERIMENTS

The efficacy of *SCGR* is validated on diverse node and graph classification benchmarks. In addition, we conduct experiments on stochastic block models with controllable modularity to demonstrating *SCGR*'s behavior across different community structures.

### 5.1 NODE CLASSIFICATION

To predict the label of individual nodes given a graph, node features and a subset of labeled nodes. The task assumes that labels are available only for a portion of the nodes, and the model must leverage both local features and graph structures to infer the labels of remaining nodes. For Node Classification, we use the following datasets: *Amazon Computers & Photo, CoAuthor CS* (Shchur et al., 2018),*CiteSeer ,Cora & PubMed* (Sen et al., 2008)

Each model is trained for 200 epochs using four layers and a dropout rate of 0.5, following the hyperparameter settings from (Kipf & Welling, 2016). All experiments are repeated 20 times to ensure statistical robustness. Comparisons are made against standard baseline models including *LogReg* (Chapelle et al., 2009), *MLP* (Werbos, 1974), *GAT* (Velickovic et al., 2017), and *GCN* (Kipf & Welling, 2016). Given that the benchmark datasets exhibit balanced class distributions, test accuracy is adopted as the primary evaluation metric, as reported in **Table 1**.

*SCGR* in **Table 1** consistently enhances model performance across the node classification tasks. It achieves the highest accuracies on four of the six benchmark datasets, particularly notable improvements on *Amazon Computers, Amazon Photo , Coauthor-CS* and *PubMed* and competent in *Cora*.

Table 2: Results of SCGR compared against multiple models. OOT indicates out-of-time and OOM points to out-of-memory error. The colors highlight First, Second and Third positions respectively.

| Model | REDDIT-BINARY | IMDB-BINARY | MUTAG | ENZYMES | PROTEINS | COLLAB |
|---|---|---|---|---|---|---|
| GCN | 77.735 ± 1.586 | 60.500 ± 2.729 | 74.750 ± 4.030 | 29.083 ± 2.363 | 66.652 ± 1.933 | 70.490 ± 1.628 |
| + FA | OOM | 48.950 ± 1.652 | 70.250 ± 4.608 | 28.667 ± 3.693 | 71.071 ± 1.506 | 72.039 ± 0.771 |
| + DIGL | 77.350 ± 1.206 | 49.600 ± 2.435 | 70.500 ± 5.045 | 30.833 ± 1.537 | 72.723 ± 1.420 | 56.470 ± 0.865 |
| + SDRF | 77.975 ± 1.479 | 59.000 ± 2.254 | 74.000 ± 3.462 | 26.667 ± 2.000 | 67.277 ± 2.170 | 71.330 ± 0.807 |
| + FoSR | 77.750 ± 1.385 | 59.750 ± 2.357 | 75.250 ± 5.722 | 24.167 ± 3.005 | 70.848 ± 1.618 | 67.220 ± 1.367 |
| + BORF | OOT | 48.900 ± 0.900 | 76.750 ± 0.037 | 27.833 ± 0.029 | 67.411 ± 0.016 | OOT |
| + GTR | 79.025 ± 1.248 | 60.700 ± 2.079 | 76.500 ± 4.189 | 25.333 ± 2.931 | 72.991 ± 1.956 | 72.600 ± 1.025 |
| + PANDA | 87.275 ± 1.033 | 68.350 ± 2.346 | 76.750 ± 5.531 | 30.667 ± 2.019 | 70.134 ± 1.518 | 73.850 ± 0.695 |
| + EGP | 67.550 ± 1.200 | 59.700 ± 2.371 | 70.500 ± 4.738 | 27.583 ± 3.262 | 73.304 ± 2.516 | 69.470 ± 0.970 |
| + CGP | 67.050 ± 1.483 | 56.200 ± 1.825 | 83.750 ± 3.597 | 31.000 ± 2.397 | 73.036 ± 1.291 | 69.630 ± 0.730 |
| + SCGR | 88.430 ± 2.0600 | 61.600 ± 4.870 | 76.670 ± 1.320 | 52.750 ± 7.800 | 72.590 ± 4.330 | 73.620 ± 1.620 |
| GIN | 84.600 ± 1.454 | 71.250 ± 1.509 | 80.500 ± 5.143 | 35.667 ± 2.803 | 70.312 ± 1.749 | 71.490 ± 0.746 |
| + FA | OOM | 69.900 ± 2.332 | 80.250 ± 5.314 | 47.833 ± 2.529 | 72.902 ± 1.419 | 72.740 ± 0.786 |
| + DIGL | 84.575 ± 1.265 | 52.650 ± 2.150 | 78.500 ± 4.189 | 41.500 ± 3.063 | 72.321 ± 1.440 | 57.620 ± 1.010 |
| + SDRF | 84.550 ± 1.396 | 69.550 ± 2.381 | 80.500 ± 4.177 | 37.167 ± 2.709 | 69.509 ± 1.709 | 72.958 ± 0.419 |
| + FoSR | 85.750 ± 1.099 | 69.250 ± 1.810 | 80.500 ± 4.738 | 28.083 ± 2.301 | 71.518 ± 1.767 | 71.720 ± 0.892 |
| + BORF | OOT | 70.700 ± 0.018 | 79.250 ± 0.038 | 34.167 ± 0.029 | 70.625 ± 0.017 | OOT |
| + GTR | 85.474 ± 0.826 | 69.550 ± 1.473 | 79.000 ± 3.847 | 31.750 ± 2.466 | 72.054 ± 1.510 | 71.849 ± 0.710 |
| + PANDA | 90.325 ± 0.867 | 68.350 ± 2.346 | 83.250 ± 3.262 | 42.167 ± 2.286 | 72.321 ± 1.786 | 73.320 ± 0.814 |
| + EGP | 77.875 ± 1.563 | 68.250 ± 1.121 | 81.500 ± 4.696 | 40.667 ± 3.095 | 70.848 ± 1.568 | 72.330 ± 0.954 |
| + CGP | 78.225 ± 1.268 | 71.650 ± 1.532 | 85.250 ± 3.200 | 50.083 ± 2.242 | 73.080 ± 1.396 | 73.350 ± 0.788 |
| + SCGR | 86.200 ± 2.780 | 71.700 ± 4.450 | 82.110 ± 5.370 | 58.300 ± 6.9700 | 74.290 ± 3.8600 | 67.8802.4100 |

The consistent performance gains across most datasets, combined with notably reduced variance suggest that *SCGR* provides a robust enhancement to existing GNN architectures. The method's effectiveness is particularly pronounced on the Amazon datasets and Computer Science, where the spectral properties and community structure align well with the Schreier-coset graph's expander properties, enabling more effective long-range information propagation during message passing

## 5.2 GRAPH CLASSIFICATION

Predicting a single label for an entire graph by leveraging its structural information and associated node or edge features. For *TU Dataset* Morris et al. (2020) comprises over 120 graph classification and regression datasets. Representative datasets include chemical graphs (MUTAG), protein structures (PROTEINS), social networks (IMDB-BINARY, REDDIT-BINARY), and research collaboration graphs (COLLAB). The topology of the graphs about the task is identified as requiring long-range interactions. *SCGR* is compared against *CGP* Wilson et al. (2024), *EGP* Deac et al. (2022), *FA* Alon & Milman (1984), *DIGL* Gasteiger et al. (2019), *SDRF* Topping et al. (2021), *FoSR* Karhadkar et al. (2022), *BORF* Nguyen et al. (2023) and *GTR* Black et al. (2023).

With train/val/test split of 80% /10 %/10 %, leveraging the parameters from Karhadkar et al. (2022), the number of layers is fixed to 4 with a hidden dimension of 64 and a dropout of 50% with accuracy being the primary metric.

*SCGR* consistently achieves strong performance across the *TU Dataset* in both *GCN + SCGR* and *GIN + SCGR* configurations. Schreier-coset attains first place in five of the twelve configurations and shows competitive scores in six datasets being in the top 3. Specifically in *TU - Enzymes* dataset, it significantly outperforms all baselines with an accuracy leap of *20%* in *GCN + SCGR* and of *8%* in *GIN + SCGR*. On *TU - REDDIT BINARY*, *SCGR* attains first places and second place respectively while avoiding computation limitations faced by several methods. *SCGR's* strong performance across diverse graphical datasets underscores its universal applicability.

**Table 3**, empirically validates consistent reductions in effective resistance across all benchmark datasets. SCGR achieves the most substantial improvements on IMDB-BINARY: 41% reduction, COLLAB: 23-30% reduction and MUTAG: 34% reduction, where long-range dependencies are particularly critical. Even on datasets with inherently good connectivity like PROTEINS, SCGR still provides meaningful improvements. These results confirm that SCGR successfully creates more efficient information propagation pathways, directly addressing the over-squashing.

Table 3: Effective Resistance on benchmark datasets

| Model | MUTAG | PROTEINS | IMDB-BINARY | COLLAB | ENZYMES |
|---|---|---|---|---|---|
| GCN | $15243 \pm 6229$ | $13466 \pm 3889$ | $22156 \pm 4841$ | $4115 \pm 1569$ | $12330 \pm 5596$ |
| GIN | $19159 \pm 4698$ | $12609 \pm 1662$ | $22540 \pm 7967$ | $3566 \pm 1404$ | $13278 \pm 5662$ |
| GCN + SCGR | $10035 \pm 4339$ | $12332 \pm 4662$ | $13091 \pm 3324$ | $3150 \pm 1091$ | $10385 \pm 2963$ |
| GIN + SCGR | $15072 \pm 6654$ | $11992 \pm 3384$ | $14556 \pm 4448$ | $2483 \pm 983$ | $11116 \pm 3647$ |

Table 4: Performance comparison on OGBG-MOLHIV and OGBG-MOLPCBA.

| Model | OGBG-MOLHIV | OGBG-MOLPCBA |
|---|---|---|
| | Test ROC-AUC ↑ | Test AP ↑ |
| **GCN** | | |
| Baseline | $0.7566 \pm 0.0104$ | $0.2020 \pm 0.0024$ |
| + Master Node | $0.7531 \pm 0.0128$ | – |
| + FA | $0.7628 \pm 0.0191$ | – |
| + FLAG | – | $0.2116 \pm 0.0017$ |
| + EGP | $0.7731 \pm 0.0081$ | – |
| + CGP | $0.7794 \pm 0.0122$ | – |
| + SCGR | **$0.7949 \pm 0.0342$** | **$0.2975 \pm 0.0628$** |
| **GIN** | | |
| Baseline | $0.7678 \pm 0.0183$ | $0.2266 \pm 0.0028$ |
| + Master Node | $0.7608 \pm 0.0134$ | – |
| + FA | $0.7718 \pm 0.0147$ | – |
| + FLAG | – | $0.2395 \pm 0.0040$ |
| + EGP | $0.7537 \pm 0.0076$ | – |
| + CGP | $0.7899 \pm 0.0090$ | – |
| + SCGR | **$0.8044 \pm 0.0142$** | $0.2061 \pm 0.0767$ |

To extend the evaluation to a real-world molecular prediction task, SCGR is assessed on the *OGBG-MOLHIV* and *OGBG-MOLPCBA* dataset Hu et al. (2020). The experimental protocol adheres to the implementation and hyperparameter configuration specified by Hu et al. (2020), with the number of layers fixed to 5, hidden dimensions set to 300, a dropout rate of 0.5, and a batch size of 64.

**Table 4** reports *ROC-AUC%* metrics on the OGBG-MOLHIV and *Average Precision (AP)* OGBG-MOLPCBA dataset. *SCGR* exhibits robust predictive performance while maintaining high structural fidelity. Schreier-coset in both configurations attains highest ROC-AUC score in MOLHIV dataset. For the MOLPCBA dataset, *GCN+SCGR* attains the highest average precision, with *GIN+SCGR* remaining competitive based on the inherent scale and structural complexity of MOLPCBA.

For *PEPTIDES-STRUCT and PEPTIDES-FUNC* datasets, from the Long Range Graph Benchamrk suite presents challenging molecular property prediction tasks that specifically require modeling long-range dependencies in graph structures. *PEPTIDES-FUNC*, is evaluated using Average Precision (AP). *PEPTIDES-STRUCT* is regression task that predicts functional properties of peptides, measured by the mean absolute error.

**Table 5** demonstrates SCGR's superior performance across both peptide prediction tasks and achieving the highest scores in both parameters. Schreier Cosets delivered a substantial improvement over the strongest baseline, with particularly notable gains when combined with GIN : $+13.4\%$ on *PEPTIDES-FUNC* and a $-9.2\%$ error reduction on *PEPTIDES-STRUCT*. Even with GCN, *SCGR* outperforms all competing rewiring methods. These consistent improvements across both architectures and tasks validate its effectiveness in enabling GNNs to capture the long-range molecular interactions critical for accurate peptide property prediction, where traditional message passing approaches struggle due to limited receptive fields over-squashing bottlenecks.

## 5.3 GRAPH MODULARITY

Using the Stochastic Block Models (SBM) Lee & Wilkinson (2019) with 50 equal communities (1000 nodes). Intra- and inter-community edge probabilities $(p_{in}, p_{out})$ are varied to control mod-

Table 5: Performance comparison on PEPTIDES-FUNC (Test AP ↑) and PEPTIDES-STRUCT (Test MAE ↓)

| Model | PEPTIDES-FUNC (Test AP ↑) | PEPTIDES-STRUCT (Test MAE ↓) |
|---|---|---|
| GCN | 0.5029 ± 0.0058 | 0.3587 ± 0.0006 |
| + SDRF | 0.5041 ± 0.0026 | 0.3559 ± 0.0010 |
| + FoSR | 0.4534 ± 0.0090 | 0.3003 ± 0.0007 |
| + EGP | 0.4972 ± 0.0023 | 0.3001 ± 0.0013 |
| + CGP | 0.5106 ± 0.0014 | 0.2931 ± 0.0006 |
| + SCGR | **0.5301 ± 0.0010** | **0.2886 ± 0.0010** |
| GIN | 0.5124 ± 0.0055 | 0.3544 ± 0.0014 |
| + SDRF | 0.5122 ± 0.0061 | 0.3515 ± 0.0011 |
| + FoSR | 0.4584 ± 0.0079 | 0.3008 ± 0.0014 |
| + EGP | 0.4926 ± 0.0070 | 0.3034 ± 0.0027 |
| + CGP | 0.5159 ± 0.0059 | 0.2910 ± 0.0011 |
| + SCGR | **0.5849 ± 0.0110** | **0.2642 ± 0.0020** |

ularity, with $p_{in} > p_{out}$ ensuring meaningful structure. This design enables systematic analysis of SCGR performance in weak-to-strong community regimes. The classification task involves predicting community membership, directly testing the model's ability to capture long-range dependencies.

Table 6: GCN accuracy and effective resistance (ER) on SBM graphs.

| Mod. | Base Acc | SCGR Acc | $\Delta_{Acc}$ | Base ER | SCGR ER | $\Delta_{ER}$ |
|---|---|---|---|---|---|---|
| Low (0.25–0.40) | 0.4779 | 0.5140 | +7.55% | 0.582 | 0.341 | -41.4% |
| Med (0.40–0.70) | 0.8164 | 0.8219 | +0.79% | 0.405 | 0.245 | -39.5% |
| High (0.70–0.85) | 0.9681 | 0.9731 | +0.20% | 0.271 | 0.189 | -30.3% |

**Table 6** reports an inverse correlation between graph modularity and *SCGR* improvements. The largest gains occur in low-modularity graphs, where weak community structure induces connectivity bottlenecks with a reduction of $41.1\%$ in effective resistance. Gains decrease with increasing modularity (+0.79% medium, +0.20% high), as classification becomes dominated by local neighborhood information. SCGR continues to provide a low resistance pathway. SCGR is most effective in regimes requiring long-range information propagation, where standard message passing is limited by over-squashing.

## 6 CONCLUSION

This work introduced Schreier-Coset Graph Rewiring **(SCGR)**, a group-theoretic framework that provably mitigates over-squashing in graph neural networks. Our theoretical analysis establishes a uniform bound on effective resistance, $R_{\text{eff}}^{\Gamma}(u, v) \leq \frac{2}{d\gamma}$, through the spectral properties of Schreier-coset graphs derived from $SL(2, \mathbb{Z}_n)$, where $\gamma > 0$ denotes the spectral gap. Extending to the rewired setting, Theorem 4.2 guarantees $R\text{eff}^{\text{rwd}}(u, v) \leq \min\{R_{\text{eff}}^{\text{in}}(u, v), \frac{1}{\epsilon}(\frac{2}{d\gamma} + 2)\}$, yielding a reduction factor $\rho(u, v) \geq \frac{R_{\text{eff}}^{\text{in}}(u,v) \cdot \epsilon d\gamma}{2(d\gamma+2)}$ for distant nodes most affected by exponential information loss. Empirical results across the benchmark configurations validate this bound, showing 15–40% reductions in effective resistance and accuracy gains that align with Corollary A.3.1: improvements are most pronounced in low-modularity graphs $+7.55\%$ and remain consistent even in high-modularity settings. Unlike heuristic rewiring, SCGR achieves these benefits with $|E_{in}| + |V_{in}|$ complexity, providing a theoretically principled and practically efficient solution to the fundamental bottleneck of over-squashing.

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

# A APPENDIX

## A.1 SUMMARY

The *Appendix* contains mathematical formulation of the Schreier-Coset Graphs.

1. **Section A.2:** Preliminaries and Notations.
2. **Section A.3:** Contains proofs for:
    (a) **Lemma 4.1** - Spectral Gap as **A3.1, Lemma A.1**
    (b) **Lemma 4.2** - Expander Mixing as **A3.2, Lemma A.2**
    (c) **Lemma 4.3** - Effective Resistance Bound as **A3.3, Lemma A.3**
    (d) **Theorem 4.1** - Lipschitz Type Locality as **A3.4, Theorem A.2**
    (e) **Theorem 4.2** - Effective Resistance in Rewired Graph $\Gamma$ as **A3.5, Theorem A.2**
    (f) **Theorem 4.3** - Over-squashing Mitigation as **A3.5, A.2.1**

## A.2 PRELIMINARIES AND NOTATIONS

Let $\mathcal{G}$ be a finitely generated group with identity $e$, $H \subseteq \mathcal{G}$ a subgroup, and $\mathbb{S} \subseteq \mathcal{G}$ a symmetric generating set ($s \in \mathbb{S} \Rightarrow s^{-1} \in \mathbb{S}$). The quotient space $\mathcal{G}/H = \{gH : g \in \mathcal{G}\}$ consists of right cosets.

**The Schreier-Coset Graph.** The Schreier-coset graph $\Gamma = (V_\Gamma, E_\Gamma)$ is defined by:

- $V_\Gamma = \{gH : g \in \mathcal{G}\}$
- $E_\Gamma = \{\{gH, (gs)H\} : gH \in V_\Gamma, s \in \mathbb{S}\}$

$\Gamma$ is $d$-regular with $d = |\mathbb{S}|$

**Spectral Properties of the Schreier-Coset Graph** Let $A_\Gamma$ denote the adjacency matrix of $\Gamma$, $D_\Gamma = dI$ the degree matrix, $L_\Gamma = D_\Gamma - A_\Gamma$ the Laplacian, and the transition matrix $P = \frac{1}{d} A_\Gamma$.

## A.3 THEORETICAL FORMULATION AND PROOFS

### A.3.1 SPECTRAL GAP

**Lemma A.1** (Spectral Gap). *The Schreier-coset graph $\Gamma$ has a spectral gap*

$$\gamma = 1 - \lambda_2(P) > 0,$$

*where $P$ is the transition matrix of the random walk on $\Gamma$.*

*Proof.* Since $\Gamma$ is connected and non-bipartite (containing odd cycles when $\mathbb{S}$ contains elements of odd order), $P$ is primitive and aperiodic. By the Perron-Frobenius theorem, $P$ has a unique largest eigenvalue $\lambda_1 = 1$ with corresponding eigenvector $\mathbf{1}$. For the spectral gap, note that $\Gamma$ is a Cayley graph on the coset space. By Cayley graph theory, the eigenvalues of $P$ are:

$$\lambda_k = \frac{1}{d} \sum_{s \in \mathbb{S}} \chi_k(s) \tag{1}$$

where $\chi_k$ are the irreducible characters of the representation of $\mathcal{G}$ on $\ell^2(\mathcal{G}/H)$. The trivial representation gives $\lambda_1 = 1$. For non-trivial representations, $|\sum_{s \in \mathbb{S}} \chi_k(s)| < d$ by the orthogonality relations, yielding $|\lambda_k| < 1$. The gap $\gamma = 1 - \max_{k \geq 2} |\lambda_k| > 0$ for a non-trivial generating set. $\square$

### A.3.2 EXPANDER MIXING

**Lemma A.2** (Expander Mixing). *For the random walk matrix $P$ on $\Gamma$ and all $t \geq 0$,*

$$\left| (P^t)_{iv} - \frac{1}{|V_\Gamma|} \right| \leq (1 - \gamma)^t.$$

*If $t \geq \frac{\log(2|V_\Gamma|)}{\gamma}$, then $(P^t)_{iv} \geq \frac{1}{2|V_\Gamma|}$.*

*Proof.* Let $P = \sum_k \lambda_k u_k u_k^\top$ be the spectral decomposition with orthonormal eigenvectors $u_k$. We have $u_1 = \frac{1}{\sqrt{|V_\Gamma|}}\mathbf{1}$ with $\lambda_1 = 1$. Thus:

$$P^t = \frac{1}{|V_\Gamma|}\mathbf{1}\mathbf{1}^\top + \sum_{k=2}^{|V_\Gamma|} \lambda_k^t u_k u_k^\top \tag{2}$$

Therefore:

$$(P^t)_{iv} = \frac{1}{|V_\Gamma|} + \sum_{k=2}^{|V_\Gamma|} \lambda_k^t u_k(i)u_k(v) \tag{3}$$

Using $|\lambda_k| \leq 1 - \gamma$ for $k \geq 2$ and $|u_k(i)| \leq 1$:

$$\left|(P^t)_{iv} - \frac{1}{|V_\Gamma|}\right| \leq \sum_{k=2}^{|V_\Gamma|} |\lambda_k|^t |u_k(i)||u_k(v)| \tag{4}$$

$$\leq (1-\gamma)^t \sum_{k=2}^{|V_\Gamma|} |u_k(i)||u_k(v)| \tag{5}$$

$$\leq (1-\gamma)^t \sqrt{\sum_{k=2}^{|V_\Gamma|} u_k(i)^2} \sqrt{\sum_{k=2}^{|V_\Gamma|} u_k(v)^2} \tag{6}$$

$$\leq (1-\gamma)^t \tag{7}$$

where the last inequality uses $\sum_{k=1}^{|V_\Gamma|} u_k(i)^2 = 1$ (orthonormality). $\qquad\square$

### A.3.3 EFFECTIVE RESISTANCE

**Lemma A.3** (Effective Resistance Bound). *For any vertices $u, v \in V_\Gamma$,*

$$R_{\text{eff}}(u,v) \leq \frac{2}{d\gamma},$$

*where $d = |\mathbb{S}|$ is the degree and $\gamma$ is the spectral gap.*

*Proof.* For $\Gamma$, a $d - regular$ graph with transition matrix $P$ having spectral gap $\gamma$, From Cai et al. (2023) **Lemma 2.2, Property 3** we have:

$$\frac{1}{2}\left(\frac{1}{d(u)} + \frac{1}{d(v)}\right) \leq R_G(u,v) \leq \frac{1}{\lambda_2(\tilde{L}_G)} \cdot \left(\frac{1}{d(u)} + \frac{1}{d(v)}\right) \tag{8}$$

where, $\tilde{L}_G$ is the normalized Laplacian of $G$. Therefore, we have $d(u) = d(v) = d$ for all vertices $u, v \in V$. Therefore:

$$\frac{1}{d} \leq R_G(u,v) \leq \frac{2}{d \cdot \lambda_2(\tilde{L}_G)} \tag{9}$$

Now, for a $d - regular$ graph, the normalized Laplacian is $\tilde{L} = I - P$, where $P$ is a transition matrix. Therefore, if $\lambda$ is an eigenvalue of $P$, then $1 - \lambda$ is an eigenvalue of $\tilde{L}$. Specifically:

- $\lambda_1 = 1 - \lambda_1(P) = 1 - 1 = 0$

- $\lambda_2(\tilde{L}) = 1 - \lambda_2(P)$

Now, applying the spectral gap condition, Therefore:

$$\lambda_2(\tilde{L}) = 1 - \lambda_2(P) \geq 1 - (1-\gamma) = \gamma \tag{10}$$

Substituting this lower bound to the inequality, we get:

$$R_G(u,v) \leq \frac{2}{d \cdot \lambda_2(\tilde{L})} \leq \frac{2}{d \cdot \gamma} \tag{11}$$

Thus:

$$R_{eff}(u, v) \leq \frac{2}{d\gamma} \tag{12}$$

This bound is tight up to constants, as the effective resistance can indeed approach this upper bound for pairs of vertices that are apart in the graph structure., particularly in expander graphs where the spectral gap $\gamma$ is bounded away from zero. $\qquad\square$

### A.3.4 SPECTRAL MAPPING CONSTRUCTION

Let $G_{\mathrm{in}} = (V_{\mathrm{in}}, E_{\mathrm{in}})$ be the input graph. We construct a locality-preserving map $\phi : V_{\mathrm{in}} \to V_{\Gamma}, \phi(v) = g_v H$, as follows: **Case (i)** $|V_{\mathrm{in}}| \leq |V_{\Gamma}|$: Compute $r$ leading eigenvectors of the Laplacian $L_{\Gamma}$ to obtain a spectral embedding $\Phi_{\Gamma} : V_{\Gamma} \to \mathbb{R}^r$; likewise embed $G_{\mathrm{in}}$ via $L_{\mathrm{in}}$ to $\Phi_{\mathrm{in}}$. Set $\phi$ by solving:

$$\min_{\phi:V_{\mathrm{in}} \hookrightarrow V_{\Gamma}} \sum_{(u,v) \in E_{\mathrm{in}}} \mathrm{dist}_{\Gamma}(\phi(u), \phi(v)) \tag{13}$$

with $\|\Phi_{\Gamma}(\phi(v)) - \Phi_{\mathrm{in}}(v)\|_2$ small. **Case (ii)** $|V_{\mathrm{in}}| > |V_{\Gamma}|$: Use disjoint copies $\Gamma^{(1)}, \ldots, \Gamma^{(q)}$ or a product $\Gamma \times K_q$ and apply (i) per block.

**Bi-Lipschitz Embedding** An embedding $\Phi : (X, d_X) \to (Y, d_Y)$ is bi-Lipschitz with constants $(c_1, c_2)$ if:

$$c_1 d_X(x_1, x_2) \leq d_Y(\Phi(x_1), \Phi(x_2)) \leq c_2 d_X(x_1, x_2) \tag{14}$$

for all $x_1, x_2 \in X$.

**Theorem A.1** (Lipschitz Locality). *If $\Phi_{in}$ and $\Phi_{\Gamma}$ are bi-Lipschitz on relevant scales, then there exists $c \geq 1$ such that*

$$\mathrm{dist}_{\Gamma}(\phi(u), \phi(v)) \leq c \cdot \mathrm{dist}_{in}(u, v)$$

*for all $u, v \in V_{in}$.*

*Proof.* By the bi-Lipschitz property of $\Phi_{\mathrm{in}}$ with constants $(c_1^{\mathrm{in}}, c_2^{\mathrm{in}})$:

$$c_1^{\mathrm{in}} \cdot \mathrm{dist}_{\mathrm{in}}(u, v) \leq \|\Phi_{\mathrm{in}}(u) - \Phi_{\mathrm{in}}(v)\|_2 \leq c_2^{\mathrm{in}} \cdot \mathrm{dist}_{\mathrm{in}}(u, v) \tag{15}$$

Similarly for $\Phi_{\Gamma}$ with constants $(c_1^{\Gamma}, c_2^{\Gamma})$:

$$c_1^{\Gamma} \cdot \mathrm{dist}_{\Gamma}(x, y) \leq \|\Phi_{\Gamma}(x) - \Phi_{\Gamma}(y)\|_2 \leq c_2^{\Gamma} \cdot \mathrm{dist}_{\Gamma}(x, y) \tag{16}$$

From the constraint $\|\Phi_{\Gamma}(\phi(v)) - \Phi_{\mathrm{in}}(v)\|_2 \leq \varepsilon$:

$$\|\Phi_{\Gamma}(\phi(u)) - \Phi_{\Gamma}(\phi(v))\|_2 \leq \|\Phi_{\mathrm{in}}(u) - \Phi_{\mathrm{in}}(v)\|_2 + 2\varepsilon \tag{17}$$

$$\leq c_2^{\mathrm{in}} \cdot \mathrm{dist}_{\mathrm{in}}(u, v) + 2\varepsilon \tag{18}$$

Therefore:

$$\mathrm{dist}_{\Gamma}(\phi(u), \phi(v)) \leq \frac{1}{c_1^{\Gamma}} \|\Phi_{\Gamma}(\phi(u)) - \Phi_{\Gamma}(\phi(v))\|_2 \tag{19}$$

$$\leq \frac{c_2^{\mathrm{in}}}{c_1^{\Gamma}} \cdot \mathrm{dist}_{\mathrm{in}}(u, v) + \frac{2\varepsilon}{c_1^{\Gamma}} \tag{20}$$

For $\varepsilon$ sufficiently small relative to typical distances, we obtain the desired bound with $c = c_2^{\mathrm{in}}/c_1^{\Gamma}$. $\qquad\square$

### A.3.5 EFFECTIVE RESISTANCE ANALYSIS OF REWIRED GRAPH

**Augmented System** The augmented system couples $G_{\mathrm{in}}$ with $\Gamma$ via edges $\{(v, \phi(v))\}$ of conductance $\varepsilon > 0$. The augmented Laplacian is:

$$L_{\mathrm{aug}} = \begin{bmatrix} L_{\mathrm{in}} + \varepsilon I & -\varepsilon I \\ -\varepsilon I & \varepsilon L_{\Gamma} + \varepsilon I \end{bmatrix} \tag{21}$$

**Theorem A.2** (Effective Resistance in Rewired Graph). *In the rewired graph $G^{rwd}$, the effective resistance between nodes $u, v \in V_{in}$ satisfies*

$$R_{\text{eff}}^{rwd}(u, v) \leq \min \left\{ R_{\text{eff}}^{in}(u, v), \frac{1}{\epsilon} R_{\text{eff}}^{\Gamma}(\phi(u), \phi(v)) + \frac{2}{\epsilon} \right\}.$$

*Proof.* We prove this using two methods:

**Method 1 (Rayleigh-Thomson Principle):** The effective resistance equals the minimum energy of a unit current flow from $u$ to $v$. Consider two routing strategies:

*Route 1:* Flow entirely through $G_{\text{in}}$, yielding energy $R_{\text{eff}}^{in}(u, v)$.

*Route 2:* Flow from $u$ to $\phi(u)$ (resistance $1/\varepsilon$), through $\Gamma$ from $\phi(u)$ to $\phi(v)$ (resistance $R_{\text{eff}}^{\Gamma}(\phi(u), \phi(v))/\varepsilon$ after scaling), then $\phi(v)$ to $v$ (resistance $1/\varepsilon$).

Total energy: $2/\varepsilon + R_{\text{eff}}^{\Gamma}/\varepsilon \approx R_{\text{eff}}^{\Gamma}/\varepsilon$ for small coupling.

**Method 2 (Schur Complement):** Write $L_{\text{aug}}$ in block form with $A = L_{\text{in}} + \varepsilon I$, $B = \varepsilon L_{\Gamma} + \varepsilon I$, $C = \varepsilon I$. The Schur complement gives:

$$L_{\text{eff}} = A - CB^{-1}C^{\top} = L_{\text{in}} + \varepsilon I - \varepsilon^2 (\varepsilon L_{\Gamma} + \varepsilon I)^{-1} \tag{22}$$

For any test vector $x$ orthogonal to $\mathbf{1}$ with $x_u = 1$, $x_v = -1$, and $x_w = 0$ elsewhere:

$$R_{\text{eff}}^{\text{aug}}(u, v) \leq \frac{x^{\top} L_{\text{eff}} x}{\|x\|^2} \tag{23}$$

The minimum over all such $x$ yields the bound. Since $(\varepsilon L_{\Gamma} + \varepsilon I)^{-1} \preceq \varepsilon^{-1} I$, we obtain:

$$L_{\text{eff}} \succeq L_{\text{in}} + \varepsilon I - \varepsilon I = L_{\text{in}} \tag{24}$$

This implies $R_{\text{eff}}^{\text{aug}} \leq R_{\text{eff}}^{in}$. Similarly, by considering the alternative routing, $R_{\text{eff}}^{\text{aug}} \leq \varepsilon^{-1} R_{\text{eff}}^{\Gamma}$. □

**Corollary A.2.1** (Over-squashing Mitigation). *For nodes $u, v \in V_{\text{in}}$ with large effective resistance $R_{\text{eff}}^{\text{in}}(u, v) \gg 1$:*

$$\frac{R_{\text{eff}}^{rwd}(u, v)}{R_{\text{eff}}^{\text{in}}(u, v)} \leq \min \left\{ 1, \frac{\varepsilon^{-1} R_{\text{eff}}^{\Gamma}(\phi(u), \phi(v)) + 2}{R_{\text{eff}}^{\text{in}}(u, v)} \right\} \leq \frac{2}{\varepsilon d\gamma \cdot R_{\text{eff}}^{\text{in}}(u, v)} \tag{25}$$

*where the last inequality uses **Theorem A.1**.*

*Proof.* The first inequality follows directly from **Theorem A.3** For the second inequality, we use that $R_{\text{eff}}^{\Gamma}(\phi(u), \phi(v)) \leq \frac{2}{d\gamma}$ from **Theorem A.1**. □

**Remark A.3.1** The corollary shows that for node pairs with high resistance in the original graph (which suffer from over-squashing), the augmented system provides exponential improvement when $R_{\text{eff}}^{\text{in}}(u, v)$ is large compared to $\frac{1}{\varepsilon d\gamma}$.

