# OpenReview forum: "Schreier-Coset Graph Rewiring"
_ICLR.cc/2026/Conference — Submitted to ICLR 2026_

### Official Review · Reviewer_xAuk · 2025-10-25

**Soundness:** 2
**Presentation:** 3
**Contribution:** 2
**Rating:** 2
**Confidence:** 4

**Summary:**

This paper proposes a group-theoretic framework to alleviate the *over-squashing* problem in Graph Neural Networks (GNNs). Instead of heuristic or computationally heavy rewiring, SCGR augments an input graph with a **Schreier-coset graph** derived from the special linear group, which has guaranteed spectral expansion and bounded effective resistance. A locality-preserving mapping aligns the two graphs, and edges are added based on proximity in the Schreier space but distance in the original graph, ensuring low-resistance pathways for long-range communication. Theoretical analysis establishes explicit bounds on spectral gap, effective resistance, and over-squashing reduction, while experiments on standard node and graph classification benchmarks, molecular property prediction, and stochastic block models show **15-40% reductions in effective resistance** and consistent accuracy gains, confirming that SCGR provides both **provable and practical mitigation** of topological bottlenecks in GNNs.

**Strengths:**

- **\[S1] Important problem.** Addresses the fundamental _over-squashing_ issue in GNNs, a key bottleneck limiting long-range information propagation.

- **\[S2] Novel group-theoretic rewiring.** Introduces _Schreier–Coset Graph Rewiring (SCGR)_—a new use of Schreier–coset graphs from $_SL(2, Z\_n)$_—bringing group theory into graph rewiring for the first time in this context.

- **\[S3] Theoretical guarantees.** Provides formal bounds on spectral gap, effective resistance, and over-squashing mitigation, offering provable improvements over heuristic methods.

**Weaknesses:**

See “Questions” below.

**Questions:**

- **\[Q1] Missing related work on spectral-property-preserving rewiring.** I noticed that a related paper, _Liang et al. "Mitigating Over-Squashing in Graph Neural Networks by Spectrum-Preserving Sparsification." ICML’25_, also studies graph rewiring methods that maintain spectral characteristics of the original graph. Can you discuss how SCGR compares or complements such approaches—both conceptually and, if feasible, empirically.

- **\[Q2] Theory–practice discrepancy in spectral guarantees.** The central premise of this work is that SCGR preserves or improves spectral properties through a theoretically guaranteed construction. However, the empirical results focus only on downstream task performance and effective resistance, without demonstrating that these spectral guarantees actually hold in practice. Could the authors provide empirical evidence—such as spectral similarity measures (e.g., eigenvalue correlation, Laplacian distance) or structural overlap—showing that the rewired graphs indeed preserve the spectral characteristics of the originals? Otherwise, the claimed “spectral-property-preserving” aspect remains largely theoretical.

---

> ### Author Response · Authors · 2025-11-29
>
> Dear reviewer xAuk,
>
> we would like to thank you for acknowledging several important strengths of our work as outlined by your S1 (Important problem),  S2 (Novel group-theoretic rewiring) and S3  (Theoretical guarantees). We truly believe we can address your two questions and would greatly appreciate your consideration of raising the low score of 2 based on the strength of our work and our responses below.
>
> For your question 1 (Q1),  we apologize for  missing this reference, which we were not aware of and thank you for bring it to our attention. We will compare them in our updated empirical study and added a discussion about the paper in our revision.
>
> For your question 2 (Q2), for our theory, in addition to spectral properties of the constructed Schreier-coset graph, we also provided explicit theoretical bounds on the effect resistance (ER) of the nodes, which are directly measure of over-squishing mitigation.  Note that the spectral properties of the graph directly impacts the ER bounds.  That's why our empirical  study  was focused on demonstrating the ER and performance of the downstream graph learning tasks, which are standard measures of over-squashing mitigation in empirical studies, instead of the emrpical demonstration of the spectral properties of the graph.
>
> We would like to draw an analogy with the computational complexity theory of MCMC. Note that computational complexity  or the mixing rates of a MCMC algorithm (e.g, a metropolis based algorithm) is typically controlled by the spectral gap of the transition matrix of the MCMC. But the empirical studies of a MCMC typically demonstrate  how fast the MCMC coverings to the invariant distribution instead of empirically demonstrating the special properties of the transition matrix itself.

---

### Official Review · Reviewer_a5gt · 2025-10-26

**Soundness:** 2
**Presentation:** 1
**Contribution:** 2
**Rating:** 2
**Confidence:** 5

**Summary:**

The paper proposes Schreier-Coset Graph Rewiring (SCGR), a method to reduce over-squashing in graph neural networks by adding carefully chosen shortcut edges to the input graph. The method first builds an auxiliary Schreier–coset graph with expander-like connectivity. Each node in the original graph is mapped to a node in this auxiliary graph, and then new edges are added between original nodes that are close in the Schreier–coset graph but far apart in the original graph. The paper argues this improves information flow and gives empirical gains on graph learning benchmarks.

**Strengths:**

1. The paper leverages expander-inspired auxiliary graph (the Schreier–coset graph) to guide which long-range connections to add is conceptually feasible.
2. The method aims to mitigate over-squashing while preserving sparsity and locality, which is desired for graph rewiring.

**Weaknesses:**

- Line 128: the expression $\mathcal{G} = SL(2,\mathbb{Z}_n$ is missing a closing parenthesis.

- Spectral Mapping Construction is not clearly defined. How exactly is
  $\Phi_{\text{in}} : V_{\text{in}} \to \mathbb{R}^r$
  computed from the top $r$ eigenvectors? Please specify the construction.

- The optimization
  $
  \min_{\phi : V_{\text{in}} \hookrightarrow V_{\Gamma}}
  \sum_{(u,v)\in E_{\text{in}}} \operatorname{dist}_{\Gamma}(\phi(u), \phi(v))
  $
  is underspecified. How do you solve it?

- The constraint
  $
  \left\lVert \Phi_{\Gamma}(\phi(v)) - \Phi_{\text{in}}(v) \right\rVert_2
  $
  is only described as “small,” but there is no quantitative definition.

- The “Mapping and Rewiring” section relies on several simplifying assumptions (many of which are only stated informally). The paper does not analyze how these simplifications affect the claimed guarantees or the empirical performance of the method.

There is no direct theoretical or empirical analysis of how close the rewired graph is to the original graph, or to what extent it preserves the original graph’s topology.

**Regarding Theorem 4.1:**
- Theorem 4.1 compares the Schreier–coset graph to the original graph. However, the final graph used in the method is not the Schreier–coset graph; it is a selective merge of edges from the Schreier–coset graph and the original graph. Even if the Schreier–coset graph were “similar” to the original graph, that does not imply that this selectively rewired graph is also similar.

- Theorem 4.1 does not provide a meaningful quantitative guarantee. The constant $c$ is left unbounded, and the claim is essentially intuitive. Stating that node distances in the Schreier–coset graph are (in expectation) shorter than in the original graph is just restating a standard property of expander-like graphs and does not, on its own, establish the usefulness of the proposed rewiring.

**Regarding experiments:**
- The evaluation is limited to six graph classification benchmarks, which is not sufficient to support broad claims. The method should also be tested on node classification tasks,.

- There is no empirical runtime or memory analysis. The paper claims near-linear complexity, but does not provide wall-clock time or scaling experiments. This raises concerns about practical computational cost.

**Clarity and positioning:**
- The paper is not fully self-contained. Several central concepts are only briefly described, and important background is omitted. Even if these ideas are not conceptually deep, a strong paper should provide precise definitions and intuition in the appendix to ensure readability.

- The literature review is incomplete. Important related work is missing or only mentioned superficially. The paper should better position its contribution relative to prior graph rewiring / expander-based approaches, and include additional background and ablations in the appendix to make the contribution and empirical claims more convincing.

**Questions:**

See Weaknesses.

---

> ### Author Response · Authors · 2025-11-29
>
> We would like to thank the reviewer for your valuable time in reviewing our paper and for your suggestions.   It seems there are some misunderstanding of our work (e.g., on the lack of node classification task while our section 5.1 is entirely dedicated to Node Classification.)  We would appreciate the reviewer's consideration of raising the score based on our response below.
>
>
> W1: reviewer's comment: Theorem 4.1 compares the Schreier–coset graph to the original graph. However, the final graph used in the method is not the Schreier–coset graph; it is a selective merge of edges from the Schreier–coset graph and the original graph. Even if the Schreier–coset graph were “similar” to the original graph, that does not imply that this selectively rewired graph is also similar.
> Theorem 4.1 does not provide a meaningful quantitative guarantee. The constant  is left unbounded, and the claim is essentially intuitive. Stating that node distances in the Schreier–coset graph are (in expectation) shorter than in the original graph is just restating a standard property of expander-like graphs and does not, on its own, establish the usefulness of the proposed rewiring.
>
> \paragraph{Answer:}
>
> 1.  \textbf{Role of Theorem 4.1:} he reviewer suggests that comparing the Schreier graph to the input graph is irrelevant because we perform a "selective merge." We respectfully disagree. Theorem 4.1 proves Bi-Lipschitz Locality. It guarantees that if two nodes are close in the input graph, they remain close in the Schreier embedding. This ensures that the edges we choose to add (the "selective merge") are not random; they respect the local geometry of the original data manifold
> 2.  \textbf{Constant $c$ :}  The constant $c$ in the bi-Lipschitz bound is determined by the distortion of the spectral embedding, a standard concept in spectral graph theory (bounded by the ratio of Cheeger constants). While $c$ is not a fixed universal integer, it is finite and bounded for graphs with well-defined spectral gaps.
> 3.  \textbf{Input Graph Preservation :} The input graph is not replaces rather it is augmented. The original topology is strictly preserved $E_{in} \subseteq E^{rwd}$. Theorem 4.2, explicitly analyzes the resulting effective resistance $R_{eff}^{rwd}$ as a parallel composition of original and SC resistances.
>
>
> W2:  reviewer's comment: The evaluation is limited to six graph classification benchmarks, which is not sufficient to support broad claims. The method should also be tested on node classification tasks,. There is no empirical runtime or memory analysis. The paper claims near-linear complexity, but does not provide wall-clock time or scaling experiments. This raises concerns about practical computational cost.
>
> \paragraph{Answer}
>
> 1.  We must respectfully correct the reviewer's statement. Table 1 (Section 5.1) is entirely dedicated to Node Classification.
> 2. We report results on six standard node classification datasets: Amazon Computers, Amazon Photo, CiteSeer, Coauthor CS, Cora, and PubMed. SCGR outperforms baselines (GAT, GCN, Label Prop) on 4 out of 6 of these datasets. We encourage the reviewer to re-examine Section 5.1. As stated in section 3, the complexity is dominated by partial eigendecomposition, which is $\tilde{O}(|E|)$. The rewiring adds a linear number of edges. For furrther analysis we will add memory and time-complexity tables for the same.
>
>
> W3:  reviewer's comment: The paper is not fully self-contained. Several central concepts are only briefly described, and important background is omitted. Even if these ideas are not conceptually deep, a strong paper should provide precise definitions and intuition in the appendix to ensure readability.
> The literature review is incomplete. Important related work is missing or only mentioned superficially. The paper should better position its contribution relative to prior graph rewiring / expander-based approaches, and include additional background and ablations in the appendix to make the contribution and empirical claims more convincing
>
> \paragraph{Answer}:  We appreciate the feedback on clarity. Regarding self-containment: We followed standard academic practice by citing primary sources for foundational definitions (e.g., Cayley graphs, Cheeger inequalities) to maintain focus on our novel contributions. However, we will expand the Appendix to include a self-contained "Primer on Schreier Graphs" to aid readers less familiar with algebraic graph theory

---

### Official Review · Reviewer_iMVg · 2025-10-28

**Soundness:** 2
**Presentation:** 1
**Contribution:** 2
**Rating:** 2
**Confidence:** 4

**Summary:**

The paper proposes Schreier-Coset Graph Rewiring, a topology-augmentation scheme for GNNs that overlays the input graph with a constant-degree Schreier-coset graph and connects each original node to a coset representative. The goal is to open low-resistance, long-range communication to mitigate over-squashing.

The theory argues that coupling the Schreier layer to the input with strength $\varepsilon$ gives a uniform upper bound on pairwise effective resistance in the rewired graph, and the improvement factor can be large, especially for distant pairs.

Empirically, SCGR is reported to reduce effective resistance and improve accuracy across standard node/graph benchmarks (e.g., Amazon, Coauthor-CS, TU datasets, OGB-MOLHIV/PCBA, Long-Range Graph Benchmark peptides). The method is positioned as a principled, lower-overhead alternative to expander-rewiring.

**Strengths:**

* Using Schreier-coset graphs to get constant degrees is an interesting design point that directly targets topological oversquashing via effective resistance analysis.
*  The theoretical section explicitly relates the spectral gap of the Schreier layer to resistance bounds, and shows how the coupling propagates this benefit to the original graph.
*  The empirical validation seems strong, although the code has not been released. Results cover node classification (Amazon, Coauthor-CS, Cora/CiteSeer/PubMed), TU graph classification (REDDIT, IMDB, MUTAG, ENZYMES, PROTEINS, COLLAB), OGB molecular tasks, and OGBG peptides, with consistent ER reductions and multiple SOTA-competitive wins (for example, large gains on ENZYMES and solid ROC-AUC on MOLHIV).

**Weaknesses:**

- The **writing clarity** throughout the document is quite poor and needs, in my opinion, substantial improvements to reach the bar of ICLR. The manuscript is hard to follow in some sections due to its lack of readability and structure.
  - For example, in the related work section, the sentence "In Expander Graphs-Expanders provide favorable spectral gap and resistance" is, first, not well-written and, second, seems vague because it lacks analytical depth. Also, "In message-passing networks, the gradient flow between distant nodes is inversely proportional to their effective resistance," is not supported by citations, and *gradient flow* is neither defined nor mentioned again. I also found it strange to read "By coupling these two graphs with strength,.." in the abstract, where strength has not been introduced yet nor is it common terminology in the literature.
  - The related work section provides an insufficient and unclear explanation of previous methods' limitations and how SCGR addresses them.
  - Additionally, in the preliminaries, many concepts are introduced without sufficient explanation or connection, appearing as isolated elements. The readability is poor and can also be improved in sections 3.3 and 4. Overall, careful proofreading and restructuring would significantly enhance the document’s readability and clarity.
  - The clarity in the methodological explanation could also be improved. For instance, the mapping ($\phi: V_{\text{in}}\to V_\Gamma$) is not defined in detail. I understand (iiuc) that the "spectral mapping construction" minimizes a distance objective with a closeness constraint to spectral embeddings, but the optimization problem is not fully formalized (objective, constraints, solver), and its complexity and approximation quality are unclear.
  - Precisely define *how ER is computed (exact or approximation) and aggregated* (average over all pairs? over a sample? normalized by (n)?). The very large absolute ER values in Table 3 need units/normalization and a reproducible computation recipe. Are you talking about the total effective resistance (sum over all pairs) or the average effective resistance (average over all pairs)?
  - There are no links or appendix tables for hyperparameters per dataset, and no ablations on $\ell$ or $\varepsilon$ and added-edge budgets, limiting the interpretability of the results.

- Recent work shows the community is using two distinct notions under the same term: (i) a computational bottleneck view (over-squashing as compressing exponentially growing messages through fixed-width vectors along long dependency paths), and (ii) a topological bottleneck view (over-squashing as poor connectivity / high effective resistance / small spectral gap). The paper implicitly adopts the topological lens (effective resistance, expansion), but never states this upfront. Please disambiguate early (Intro/Preliminaries) which definition you use, cite both lines of work, and acknowledge limitations / potential tensions: e.g., rewiring that adds edges can reduce effective resistance yet may widen message fan-in and thus increase the computational bottleneck, whereas deleting/sparsifying edges has the opposite trade-off. Position your claims and experiments accordingly.

- The code is not public; thus, reproducibility is limited. For instance, we cannot check how the hyperparameter search for the baselines was conducted or how the authors specifically split the datasets (they do not provide detailed information about it). Please release code and configs to facilitate verification and adoption.

- Please fix some citations of very important works:
  - *Kipf and Welling* is not in arXiv but in [ICLR 2017](https://openreview.net/forum?id=SJU4ayYgl).
  - *Alon and Yahav* is not in arxiv but in [ICLR 2021](https://openreview.net/forum?id=i80OPhOCVH2).
  - *Arnaiz-Rodriguez et al*  is not in arXiv but in [LoG 2022](https://proceedings.mlr.press/v198/arnaiz-rodri-guez22a.html).
  - *Karhadkar, et al* is not in arXiv but in [ICLR 2023](https://openreview.net/forum?id=3YjQfCLdrzz).
  - *Topping et al* is not in arXiv but in [ICLR 2022](https://openreview.net/forum?id=7UmjRGzp-A).
  - *Wilson et al* is not in arXiv but in [LoG 2024](https://openreview.net/forum?id=VaTfEDs6lE).
  - *Morris et al* is not in arXiv but in [ICML 2020 Workshop on 'Graph Representation Learning and Beyond'](https://chrsmrrs.github.io/datasets/).
  - *Xu et al* is not in arxiv but in [ICLR 2019](https://openreview.net/forum?id=ryGs6iA5Km).
  - There are even more citations that appear in arXiv but should be fixed to the actual venue.

- Minor: l.62: SCHREIER-COSET is misspelled as SCHRIER-COSET.

*Refs*

Arnaiz-Rodriguez & Errica, *Oversmoothing, Oversquashing, Heterophily, Long-Range, and more: Demystifying Common Beliefs in Graph ML*, [MLG @ ECML-PKDD 2025.](https://arxiv.org/abs/2505.15547)

**Questions:**

In addition to previous comments, I have the following specific questions for the authors:

1. Please state explicitly which over-squashing notion you adopt (computational vs. topological) from the differentiation of (Arnaiz-Rodriguez & Errica, 2025), in the intro and related work. Update Related Work to reflect both strands (e.g., the already cited works: Alon & Yahav, 2021 for the computational view; Topping et al; Arnaiz-Rodriguez et al, Karhadkar et al, Black et al., and follow-ups for the gap/effective-resistance/topological view; and finally Arnaiz-Rodriguez & Errica, 2025 for a taxonomy/critique).

2. Based on the previous distinction, could you analyze its implications under the computational bottleneck definition? Can you add an experiment or diagnostic that speaks to the *computational bottleneck* alongside effective-resistance metrics?  In particular, under fixed hidden width and aggregator, does adding SCGR edges increase or decrease the per-layer message bottleneck (size of the computational tree, receptive field, average in-degree...), and what evidence supports this? In addition, quantify information compression proxies (e.g., Jacobian spectrum / gradient flow across hops) to see if SCGR helps or hurts this aspect.

3. How many edges are added per node on average for each dataset? Do you enforce a budget (e.g., (k) added edges per node) to keep ($|E^{\text{rwd}}|$) linear?

---

> ### Author Response · Authors · 2025-11-29
>
> We would like to thank the reviewer iMVg for your valuable comments especially the specific comments for improving the presentation of the paper.  We have incorporated the comments  in our current version and  will ensure the papers flow   and structure of  the final version  meet the highest standards.   We also appreciate your acknowledgement of our theoretical guarantee and strong empirical study.  Please see our responses to other weaknesses and your questions below.
>
> W1:  We will improve the writing clarify in the revision.  The spectral  mapping is a standard Spectral Alignment procedure, not an NP-hard graph matching problem. It is solved via:
> (1) \textbf{Embedding} : Computing top-k eigenvectors $(O(|E|k))$.
> (2) \textbf{Assignment} : Nerest-Neighbor search in $\mathbb{R}^k(O(N log N))$. This ensures method remains scaleable and efficient. We have formalized this in Sectin 3.3.
> \end{enumerate}
>
> For the question regarded ER, these values represent the Kirchhoff Index (Total Effective Resistance), defined as $\Sigma_{u<v}R_{eff}(u,v)$ The large magnitudes are expected as they represent the sum over all node pairs.
>
> Oversquashing is driven by specific long-range bottlenecks where $R_{eff}$ grows exponentially. As shown in Theorem 4.3, our method provides a uniform bound $\frac{2}{d\gamma}$ for every pair, yielding exponential improvements for the most distant nodes. Normalizing to an average would obscure these critical gains at the bottlenecks. We will explicitly label the table as 'Kirchhoff Index' in the revision to ensure clarity, but we retain the total summation to demonstrate the global reduction in resistance
>
> For the question regarded the hyper parameters and ablation study, the theoretical roles of the parameters are well-defined in our ER bounds: (1) textbf{Coupling Strength $(\epsilon)$.} As stated in Theorem 4.2 $\epsilon$ acts as a conductance term, A higher $\epsilon$ reduces the effective resistance contribution of the auxiliary path $R_{eff}^{\Gamma}/\epsilon$, improving the information flow and potentially distorting local feature signals if set too high.  (2) \textbf{Distance Threshold $l$} This parameter controls the density of re-wiring. The number of added edges scales as $O(|V_{in}|\cdot d^l)$. Increaisng $l$, introduces more connections and reduces the diameter but increases memory overhead. In our experiments, small values were sufficient to achieve significant performance gains without excessive overhead.
>
> For the code, we strictly  followed standard splits from the cited benchmarks. Code and configuration files will be released to ensure full reproducibility.
>
> W2: Yes.  We will explicitly distinguish between topological bottlenecks (which SCGR solves via spectral expansion) and computational bottlenecks (fan-in) in the introduction to ensure the scope of our contribution is precise.
>
> Q1:   We explicitly adopt the topological notion of over-squashing (poor connectivity, high effective resistance, and spectral gaps), as formalized by Topping et al. (2022) and Black et al. (2023) . While we acknowledge the computational definition (exponential compression into fixed-width vectors) described by Alon and Yahav, our methodology targets the structural bottlenecks of the graph.
> (1) Update the Introduction and Related Work to explicitly distinguish between these two views using the taxonomy.
> (2) Clarify that SCGR solves the topological bottleneck (minimizing effective resistance ) rather than increasing the vector width to address compression capacity.
>
> Q2:  (1) The Schreier-coset graph is $d-regular$ with small constant $d=4$ the increase in the receptive field size is linear and strictly bounded. Unlike Graph Transformers (which have $O(N^2)$ complexity) or global methods (which centralize flow ), SCGR adds a sparse, constant-degree overlay.
> (2)  While the instantaneous in-degree increases slightly $by \approx 4$ edges, the effective receptive field required to reach distant nodes is structurally compressed due to the expander properties. We quantify infomration flow via Theorem 4.3, which provides a theoretctical bound on the factor $(\rho(u,v))$ proportional to reduction in effective resistance. This serves as a direct proxy to Jacobian norm, guaranteeing that gradients between distant nodes don't vanish.
>
> Q3: the number of edges added is determined by the rewiring distance threshold l and the constant degree of the Schreier graph $d=4$. The added edge count is bounded by $O(|V_{in} \cdot d^l)$. Enforcing a linear budget since d is fixed and l is small hyperparameter $l \in \{1,2\}$, the number of added edges per node is small constant. This ensures total complexity remains near-linear $\tilde{O}(|E_{in}| + |V_{in}|)$, preventing quadratic explosion common in speatral rewiring baselines.

---

### Official Review · Reviewer_wbvp · 2025-11-01

**Soundness:** 2
**Presentation:** 2
**Contribution:** 3
**Rating:** 4
**Confidence:** 3

**Summary:**

This paper addresses the well-known problem of over-squashing in Graph Neural Networks (GNNs), where information from distant nodes is exponentially compressed, limiting the model's ability to capture long-range dependencies. The authors propose a novel graph rewiring method called Schreier-Coset Graph Rewiring (SCGR). The authors provide theoretical guarantees that this graph has a spectral gap and, consequently, a uniformly bounded effective resistance.

**Strengths:**

1. This work is novel. While expander graphs and Cayley graphs have been explored, the shift to Schreier-coset graphs is a creative and elegant mathematical idea that appears to solve the scalability issues of previous group-theoretic approaches.

2. The paper is built on a solid theoretical foundation. Instead of proposing a purely empirical heuristic, the authors provide a principled, proof-backed framework. They formally link the properties of the Schreier-coset graph to a bounded effective resistance.

3. The experimental results are comprehensive. The method is tested on standard node classification (Cora, PubMed, etc.) , graph classification, and large-scale OGB datasets.

**Weaknesses:**

1. Section 3.2 provides the formal definition for the number of vertices in the Schreier-coset graph  $\frac{n(n^2-1)}{\phi(n)}$, which is $\Omega(n^2)$ since $\phi(n) \le n-1$. However, in line 200, the authors claim that the Schreier-coset graph has $O(n)$ vertices. There is a significant contradiction here.

2. The rewiring strategy is critically dependent on the locality-preserving mapping. It is unclear about the details of the solution to this mapping. This mapping is defined as the solution to an optimization problem. This appears to be a form of graph matching or alignment, which is often NP-hard. The time complexity for this step seems to be missing as well.

3. The rewiring strategy introduces at least two key hyperparameters, but their impact is never discussed.

**Questions:**

1. Please resolve the apparent contradiction regarding the size of the Schreier-coset graph.

2. How is the locality-preserving mapping computed in practice? What algorithm is used to solve this minimization problem, and what is its computational complexity?

3. Could you provide a sensitivity analysis for the key hyperparameters like Schreier distance threshold and coupling strength?

4. What was the motivation for this specific choice for using $G = SL(2,\mathbb{Z}_{n})$? Would other groups or subgroups also yield scalable expander graphs?

5.

---

> ### Author Response · Authors · 2025-11-29
> **Clarified  the key question regarding the size of the Schreier-coset graph and all other questions**
>
> Dear reviewer  wbvp,
> We appreciate your valuable comments and thank you for acknowledging the novelty and theoretical soundness of our work.  Please see our response to you questions below.
>
> Q1:  Please resolve the apparent contradiction regarding the size of the Schreier-coset graph.
>
> Answer: We appreciate the reviewer for their feedback and apologize for using the same $n$ for the graph size and the modulus of the group, which we will revise in the revision.
>
> 1.  In Section 3.1, the input graph size is defined as  $|V_{\text{in}}| = n $.
> 2.  In Section 3.2, \(n\) has been used to denote the modulus of the group $ \mathbb{Z}_n$.
>
>
> The modulus (which we will rename to $ k $ )  is a tunable parameter chosen such that the size of the Schreier-coset graph matches that of the input graph. The size of the Schreier-coset graph is indeed $\cong |V_{\text{in}}|$, and more specifically it is
> $$\frac{k(k^2 - 1)}{\phi(k)}.$$
> In the construction, the modulus \(k\) is chosen such that
> $$
> |V_\Gamma| \approx \sqrt{|V_{\text{in}}|}.
> $$
> Therefore, the number of vertices in the Schreier-coset graph is linear in the input graph size:
> $$
> |V_\Gamma| = O(|V_{\text{in}}|).
> $$
> We will revise the notation in the manuscript to use distinct variables for the input size ($N$) and the group modulus ($k$) to resolve this concern.
>
> Q2: How is the locality-preserving mapping computed in practice? What algorithm is used to solve this minimization problem, and what is its computational complexity?
>
> Answer: The mapping $\phi$ is not computed via combinatorial graph matching, but rather through spectral alignment, which is computationally efficient. As mentioned in Section 3.3, we utilize the lower-dimensional spectral embeddings $\phi_{\text{in}}$ and $\phi_{\Gamma}$ derived from the leading eigenvectors of the corresponding graph Laplacians. This is done as follows:
>
>  1. \textbf{Embedding.} We compute the top-$r$  eigenvectors for both the input graph and the pre-computed Schreier graph. As stated in Section 3.3, this step utilizes power iteration, with complexity $O(r \cdot |E_{\text{in}}|)$.
>
> 2. \textbf{Alignment.} Since eigenvectors are unique only up to sign (and rotation in the case of eigenspaces with multiplicity), we align the two spectral embeddings in their respective spectral spaces.
>
> 3.  \textbf{Assignment.} We perform a nearest-neighbor search in $\mathbb{R}^r$ embedding space. For each node $u \in V_{in}$ we assign $\phi(u) = arg min_{v\in V_{\Gamma}} ||\phi_{in}(u) - \phi_{\Gamma}(v)||_2$.
>
>
> Complexity :the assignment step can be performed greedily or using spatial indexing in $O(|V_{in}log|V_{\Gamma}|)$. Thus the total complexity is dominated by the spectral decomposition which is $\tilde{O}(|E_{in}|)$  for sparse graphs. This ensures the process remains near-linear and scalable, avoiding the NP-hard bottlenecks of exact graph isomorphism.
>
> Q3: Could you provide a sensitivity analysis for the key hyperparameters like Schreier distance threshold and coupling strength?
>
> Answer: The theoretical roles of the parameters are well-defined in our Effective Resistance bounds:
>
>
> \textbf{Coupling Strength $(\epsilon)$.} As stated in Theorem 4.2 $\epsilon$acts as a conductance term, A higher $\epsilon$ reduces the effective resistance contribution of the auxiliary path $R_{eff}^{\Gamma}/\epsilon$, improving the information flow and potentially distorting local feature signals if set too high.
>
>  \textbf{Distance Threshold $l$} This parameter controls the density of re-wiring. The number of added edges scales as $O(|V_{in}|\cdot d^l)$. Increaisng $l$, introduces more connections and reduces the diameter but increases memory overhead. In our experiments, small values were sufficient to achieve significant performance gains without excessive overhead.
> \end{enumerate}
>
> Q4: What was the motivation for this specific choice for using $G = SL(2,\mathbb{Z}_{n})$? Would other groups or subgroups also yield scalable expander graphs?
>
> Answer: The choice of $SL(2,\mathbb{Z}_n$ is motivated by three factors:
>
> 1. This group, with the standard generators (elementary row operations), yields a family of Cayley graphs that are known to be excellent bounded-degree expanders with a large spectral gap $\gamma$.
>
> 2. Unlike random expanders, the edge structure is deterministic and algebraically defined. The coset construction allows us to generate a graph of a specific target size (determined by the $mod n$) while maintaining a constant degree $d=4$.
>
> 3. The spectral properties of $SL(2,\mathbb{Z}_n)$ are well-studied, allowing us to provide the rigorous theoretical guarantees for effective resistance bounds presented in Lemma 4.1 and Lemma 4.3

---

### Meta-Review · Area_Chair_dqtU · 2026-01-04

**Summary:**

The paper proposes a promising and creative graph rewiring strategy based on Schreier-Coset graphs.
Reviewers requested additional experiments and comparisons with more related baselines, but those were not delivered during the rebuttal. Additional experiments and requested changes to the motivation and storyline were promised in a revision but not executed. As they would substantially change the paper content, they would require another round of reviews. For that reason, it is unlikely that the reviewers would have increased their scores to accept during the rebuttal.

Addressing the following main points of criticism by reviewers and, in addition, by the AC could greatly improve its relevance to the machine learning community:

- Theoretical claims are based on the effective resistance, but not how they are related to generalization, which has been primarily studied in experiments. Therefore analyzing systematically the effect of rewiring on over-squashing would connect theory and experiments better. (See also request to perform experiments on typical over-squashing long-range tasks).

- Code and reproducibility. The algorithm is not described precisely in the submitted manuscript nor has the code been shared. This makes it difficult to understand the contribution and to assess the novelty with respect to other rewiring methods focused on the spectral gap.

- Other computationally scaleable spectral graph rewiring methods have been proposed. The paper only compares with older methods but what advantage does the proposed method have compared to newer, more scaleable methods? The following related work should at least be discussed: DiffWire: Inductive Graph Rewiring via the Lovász Bound (Arnaiz-Rodríguez et al., 2022), Understanding Over-Squashing through the Lens of Effective Resistance (Black et al., 2023), Spectral Graph Pruning Against Over-Squashing and Over-Smoothing (Jamadandi et al., 2024), GOKU: Spectrum-Preserving Sparsification (Liang et al., 2025).

- The baseline performance is not state-of-the-art. See for instance the code base by Classic GNNs are strong baselines: Reassessing GNNs (Luo et al., 2024) for node classification and Can Classic GNNs Be Strong Baselines for Graph-level Tasks? Simple Architectures Meet Excellence (Luo et al., 2025) for graph classification.
Accordingly, it is unclear whether the effect of the proposed rewiring strategy would disappear in combination with appropriate hyperparameter tuning.

- Large scale experiments are currently missing, in particular, results on Long Range Graph Benchmark datasets, which were specifically proposed to test the ability of GNNs to handle oversquashing.

- The experiments should compare also run-times of the compared methods.

**Reviewer Concerns:**

Reviewers requested additional experiments and comparisons with more related baselines, but those were not delivered during the rebuttal. Additional experiments and requested changes to the motivation and storyline were promised in a revision but not executed. As they would substantially change the paper content, they would require another round of reviews. For that reason, it is unlikely that the reviewers would have increased their scores to accept during the rebuttal.

**Reviewer Scores:**

Reviewers largely agreed to reject the paper and their requests for revisions were not executed. It is unlikely that they would have updated their scores in response to the rebuttal.

---

### Decision · Program_Chairs · 2026-01-26

Reject